# The Anti-Inflammatory and Skin Barrier Function Recovery Effects of *Carica papaya* Peel in Mice with Contact Dermatitis

**DOI:** 10.3390/ijms262211122

**Published:** 2025-11-17

**Authors:** Seonah Park, Kyoungmin Sun, Yeojin Kim, Hyorhan Son, Jimi Lee, Soyeon Kim, Hyungwoo Kim

**Affiliations:** Division of Pharmacology, School of Korean Medicine, Pusan National University, Yangsan 50612, Republic of Korea

**Keywords:** *Carica papaya*, contact dermatitis, skin barrier function, inflammation

## Abstract

With growing concerns over the safety of synthetic substances, the development of plant-derived alternatives with minimal adverse effects has gained significant attention. *Carica papaya* L. peel contains a rich profile of bioactive compounds, including papain, flavonoids, and vitamin C, which exhibit potent antioxidant and anti-inflammatory properties. This study aimed to evaluate the effects of an ethanol extract of *C. papaya* peel (EECP) on inflammation and skin barrier dysfunction in a mouse model of contact dermatitis (CD) induced by 1-fluoro-2,4-dinitrofluorobenzene (DNFB). Mice were treated by applying EECP at three different levels (60, 80, and 600 μg) to dorsal skin for six days. Skin lesion severity, skin color, skin barrier function (SBF, as indicated by water content and water-holding capacity (WHC)), histopathological abnormalities, cytokine levels, filaggrin and Intercellular Adhesion Molecule-1 (ICAM-1) expression, and phosphorylation of MAPK (Mitogen-Activated Protein Kinase) signaling molecules were assessed. EECP treatment significantly alleviated the CD-associated dermal symptoms induced by DNFB, including skin fissures, scabbing, roughness, changes in color, water content, and WHC, as well as petechiae. EECP also prevented histopathological abnormalities such as epidermal hyperplasia, spongiotic changes, and immune cell infiltration. In addition, EECP suppressed the production of pro-inflammatory cytokines, viz. TNF-α, IFN-γ, IL-6, and MCP-1. In addition, EECP restored filaggrin expression and inhibited ERK (Extracellular signal-regulated kinases) phosphorylation and ICAM-1 expression in HaCaT cells. In summary, *C. papaya* peel demonstrated therapeutic potential by effectively suppressing inflammation and restoring SBF. These findings support the potential use of EECP as a safe and effective botanical candidate for the treatment of CD and the promotion of overall skin health

## 1. Introduction

Interest in skin health has increased considerably during recent years. Skin condition affects not only appearance but also mental health, social interactions, and even economic status, and individuals with a skin disease frequently experience depression, social isolation, reduced quality of life, and financial difficulties [1]. Consequently, the demand for safe and effective skincare strategies continues to grow. However, synthetic chemicals commonly used in cosmetics pose significant health risks, such as allergies, sensitivities, and skin irritation, as well as environmental harm. Furthermore, prolonged use can cause hormone disruption and carcinogenic effects. Thus, cosmetics formulated with naturally derived ingredients are gaining attention as safer and more sustainable alternatives [2].

*Carica papaya* L. is rich in bioactive compounds, including flavonoids and sterols, which have demonstrated anti-inflammatory, anti-arthritic, antimicrobial, and wound-healing activities in preclinical studies. These findings suggest that papaya extracts may offer therapeutic potential for inflammatory conditions. However, clinical evidence and standardized formulations remain limited, highlighting the need for further research to establish their efficacy and safety [3,4,5].

Specifically, papaya peel contains high levels of vitamin C, phenols, and flavonoids and has excellent antioxidant activity [6]. In addition, cysteine protease activities, including those of papain and chymopapain, are ~1.6 times higher in papaya peel than in papaya latex [7]. Consequently, papaya peel, which is typically discarded, has a higher bioactive compound content and is more accessible than pulp, and thus, represents a valuable natural resource [8].

The stratum corneum (SC) serves as a critical barrier by protecting the body against external irritants, preventing the ingress of harmful substances, and minimizing moisture loss. Skin surface hydration is a key indicator of skin barrier function (SBF). Under dry environmental conditions, reduced SC hydration compromises SBF, leading to skin damage characterized by cracking, fissuring, and chapping [9]. Impaired SBF facilitates allergen penetration, resulting in immune responses, inflammation, and itching, and creates a vicious cycle closely linked to various skin diseases [10]. Contact dermatitis (CD) patients exhibit skin barrier dysfunction and clinical symptoms proportional to the degree of SBF impairment. Dry conditions have been reported to increase the incidence and severity of CD, whereas short-term exposure to high humidity can alleviate symptoms [11]. Therefore, SBF restoration through moisturization and appropriate humidity management is essential for preventing and treating skin diseases like CD [12].

CD is a common inflammatory skin disorder that affects a significant proportion of the population [13]. The hallmark histopathological features of CD are spongiosis, immune cell infiltration, and vasodilation [14], which are closely associated with clinical manifestations, such as erythema and pruritus. In addition, symptoms like fissuring, scabbing, and roughness may occur depending on the location of lesions and the disease phase [15].

Given this background, this study aimed to evaluate the ability of *C. papaya* peel to ameliorate CD symptoms and improve SBF. The experimental parameters considered included skin lesion severity, skin color measurements of erythema, melanin indices, skin hydration status (water content and water-holding capacity (WHC), cytokine levels, and histopathological abnormalities in inflamed tissues in a DNFB (1-fluoro-2,4-dinitrofluorobenzene) induced mouse model of contact dermatitis (CD).

## 2. Results

### 2.1. EECP Improved CD Symptoms and Reduced Skin Lesion Scores

Topical application of 0.1% DNFB to mouse dorsal skin induced characteristic features of CD, including petechia, erythema, fissuring, scabbing, and skin roughness, and these symptoms were alleviated by EECP (Figure 1A), Furthermore, EECP application significantly reduced skin lesion scores (Figure 1B). However, although EECP did not significantly affect skin weight, DEX elicited significant reductions (Figure 1C).

### 2.2. EECP Alleviated Erythema and Melanin Indices

In the CTL group, pronounced flushing was observed on dorsal skin with an erythema index nearly 5-fold that of the NOR group (*p* < 0.001). Topical application of EECP at 180 or 600 μg/day significantly reduced erythema indices (*p* < 0.05) (Figure 2A). In addition, melanin indices were significantly reduced in all EECP-treated groups (*p* < 0.001) (Figure 2B). Erythema and melanin indices were also reduced in the DEX group (Figure 2).

### 2.3. EECP Increased Skin Water Content and Water-Holding Capacity

The water content in the CTL group was reduced to approximately one-third of the normal level, while EECP treatment resulted in a significant and dose-dependent increase (Figure 3A). At the 90 s after removing wet gauze, WHC in the CTL group declined to nearly half of normal levels, whereas all EECP-treated groups showed significant improvements (*p* < 0.001) (Figure 3B).

### 2.4. EECP Effectively Inhibited Epidermal Hyperplasia and Immune Cell Infiltration of Inflamed Tissues

DNFB exposure induced histopathologic abnormalities, including epidermal hyperplasia and spongiotic changes in the basal layer (Figure 4A). In the CTL group, epithelial thickness increased ~2-fold, and the number of infiltrating immune cells was ~6-fold greater than in the NOR group (Figure 4A). EECP treatment significantly reduced epithelial thickness and infiltrating immune cell numbers at all tested concentrations. DEX administration also resulted in significant improvements in these histological features (Figure 4B,C).

### 2.5. EECP Reduced the Production of TNF-α, IFN-γ, IL-6 and MCP-1 in Inflamed Tissues

Repeated DNFB exposure significantly elevated the levels of four pro-inflammatory cytokines, viz., TNF-α, IFN-γ, IL-6, and MCP-1 (*p* < 0.001). Topical administration of EECP at 180 μg/day significantly reduced TNF-α and IL-6, while 600 μg/day markedly suppressed all four cytokines (Figure 5).

### 2.6. EECP Inhibited TNF-Induced Pro-Filaggrin Accumulation and Phosphorylations of ERK and P38, and Restored Filaggrin Expression in HaCaT Cells

EECP treatment dose-dependently suppressed the TNF-induced pro-filaggrin accumulation in HaCaT cells, while concomitantly increasing the expression of filaggrin (Figure 6A). In addition, EECP treatment inhibited TNF-induced phosphorylation of ERK (Extracellular signal-regulated kinases) and P38 in HaCaT cells, while having no significant effect on JNK (c-Jun N-terminal kinase) phosphorylation (Figure 6B).

### 2.7. EECP Inhibited TNF-α-Induced ICAM-1 and Cytokine Expression in HaCaT Cells

EECP treatment effectively suppressed the TNF-α-induced upregulation of ICAM-1 mRNA in HaCaT cells (Figure 7A). EECP also significantly reduced the relative mRNA expression levels of IL-6, IL-8, and MCP-1 in HaCaT cells.

## 3. Discussion

Topical therapies for dermatitis mainly include corticosteroids, calcineurin inhibitors, and phosphodiesterase-4 (PDE4) inhibitors, each acting through distinct inflammatory pathways. Corticosteroids remain the most widely used agents, but their long-term use can lead to adverse effects such as skin atrophy, telangiectasia, and barrier impairment [16,17]. Calcineurin inhibitors, including tacrolimus and pimecrolimus, offer steroid-sparing alternatives by suppressing T-cell and mast cell activation and cytokine secretion; however, local irritation and potential systemic risks limit their use [18]. PDE4 inhibitors, recently introduced as non-steroidal anti-inflammatory drugs, also reduce inflammatory cytokine expression but are often restricted by cost and local side effects [19].

Given these limitations, increasing attention has been directed toward plant-derived natural products with anti-inflammatory and barrier-protective properties. The present study shows that the ethanolic extract of papaya peel improves skin water retention and enhances WHC, and thus, helps prevent further SBF breakdown and highlights the therapeutic potential of the extract.

The skin barrier is composed of the stratum corneum, and its structural and functional integrity is highly dependent on water homeostasis through moisture retention. Since the majority of enzymes that facilitate essential SC activities are hydrolytic, hydration at below threshold levels reduces enzymatic efficiency [20,21]. In this study, EECP-treated groups exhibited significantly higher skin water contents and WHCs than DNFB-treated controls (Figure 3A,B), suggesting that EECP application effectively restored diminished moisture levels. Notably, DEX treatment failed to significantly improve WHC after 90 s, while EECP at all concentrations achieved significant improvements. SBF improvements were also associated with the visible amelioration of clinical symptoms such as skin roughness and fissuring [22]. In the present study, visual assessment revealed EECP also improved skin roughness and reduced fissures (Figure 1A), suggesting that EECP might alleviate the cutaneous symptoms of CD by enhancing SBF.

CD typically presents with symptoms of erythema, swelling, exudation, scabbing, papules, and vesicles. In patients with dermatitis, the skin typically shows erythema during the acute phase, and as the condition progresses to a chronic stage, post-inflammatory hyperpigmentation develops, leading to skin darkening. [15]. In the present study, EECP significantly improved clinical symptoms of erythema, petechiae, skin roughness, scabbing, and fissuring in our murine model (Figure 1A,B), and significantly reduced DNFB-induced increases in erythema and melanin indices (Figure 2A,B). The papaya-derived components in EECP probably caused these therapeutic effects. Notably, papain is a proteolytic enzyme derived from papaya that has been shown to alleviate clinical skin lesions, such as dryness, edema, and excoriation [23], and caffeic acid (another component of papaya) is known to reduce skin erythema by inhibiting the NF-κB and AP-1 (activator protein 1) signaling pathways [24].

Epidermal thickening is a hallmark of CD and results from skin barrier disruption followed by keratinocyte hyperproliferation and abnormal differentiation, including parakeratosis [25]. TNF-α and IFN-γ, key Th1 cytokines, contribute to epidermal hyperplasia by activating keratinocytes, promoting the release of pro-inflammatory mediators, and impairing SBF [16]. In this study, EECP treatment significantly prevented DNFB-induced epidermal thickening (Figure 4A,B) and suppressed the expression of IFN-γ and TNF-α (Figure 5A,B). Additionally, improvements in skin lesions (Figure 1A,B) and restoration of skin barrier function (Figure 3A,B) were also observed in EECP-treated groups. These findings collectively suggest that EECP may restore SBF by suppressing inflammatory cytokines, including IFN-γ and TNF-α, thereby improving skin lesions and preventing epidermal thickening.

IL-6 contributes to immune cell infiltration by enhancing the migration of neutrophils and macrophages to inflamed tissues through binding to IL-6Rα [26]. MCP-1, a key inflammatory chemokine, is also elevated in CD and plays a critical role in the recruitment and activation of immune cells [27,28]. In this study, EECP significantly reduced IL-6 and MCP-1 levels (Figure 5C,D) and suppressed immune cell infiltration (Figure 4C). These findings suggest that EECP suppresses immune cell infiltration by downregulating the levels of cytokines and chemokines. Moreover, spongiotic change, which is frequently associated with lymphocytic exocytosis [29], was clearly observed in the DNFB control group but was barely detected in the EECP-treated groups (Figure 4A). This suggests that EECP may alleviate histopathological features, such as spongiosis, by inhibiting immune cell infiltration.

A recent study reported that the ethyl acetate fraction of papaya leaf extract, containing quercetin and kaempferol glycosides as well as carpaine, inhibited the production of inflammatory mediators, including IL-1β and IL-6, in RAW 264.7 cells [30]. These findings suggest that similar anti-inflammatory effects may also apply to papaya peel extract.

Papaya contains bioactive carotenoids, with lycopene being particularly notable for its strong antioxidant properties. These carotenoids scavenge reactive oxygen species (ROS), modulate immune responses, and reduce the expression of key inflammatory mediators such as IL-6 and MCP-1 [31], consistent with the findings of the present study. They also inhibit the phosphorylation of MAPK family members, thereby suppressing inflammation and collagen degradation [32]. In addition, flavonoids, including apigenin, fisetin, luteolin, naringenin, and quercetin, exhibit anti-inflammatory effects by inhibiting the production of TNF-α, IL-6, IL-8, IFN-γ, and MCP-1 [33]. They also suppress melanin synthesis by downregulating melanogenesis-related proteins such as TRP-1 and TRP-2 [34]. The above components likely contributed predominantly to the cytokine-suppressing anti-inflammatory effects observed in this study.

In vitro experiments using HaCaT keratinocytes were designed to recapitulate the inflammatory environment observed in contact dermatitis. Direct stimulation with DNFB is not suitable for keratinocytes in culture, as it primarily acts as a hapten in vivo. Therefore, TNF-α, a key pro-inflammatory cytokine elevated in inflamed skin, was used to induce an inflammatory response in HaCaT cells, allowing the assessment of anti-inflammatory effects under controlled conditions.

Filaggrin, an important natural moisturizing factor, can have its expression suppressed by pro-inflammatory cytokines. Studies have shown that TNF-α, IL-4 and IL-13 inhibit filaggrin expression and further weaken skin barrier function [35,36]. In our study, topical EECP restored reduced filaggrin expression by TNF-α in HaCaT cells (Figure 6A), which suggests that EECP may influence SBF by restoring filaggrin expression levels through the suppression of TNF-α.

Vitamin C, a major component of papaya peel, supports skin health by promoting keratinocyte differentiation, reducing oxidative stress, and enhancing cornified envelope formation, leading to a more compact stratum corneum. These effects help maintain the skin barrier, reduce trans-epidermal water loss (TEWL), and improve inflammatory skin conditions such as dermatitis. Additionally, vitamin C stimulates dermal collagen production, further enhancing skin structure and resilience [37]. Considering the above results, vitamin C is expected to be one of the key components contributing to the observed improvement in SBF in this study.

Phospho-ERK was highly expressed in both mouse and human AD skin, and topical application of an ERK inhibitor alleviated clinical symptoms, histological alterations, increased TEWL, and the decreased expression of filaggrin in the AD-like NC/Nga murine model [38]. In addition, a p38 inhibitor also alleviated the reduction in filaggrin expression induced by IL-17 stimulation [39]. As shown in Figure 6B, EECP inhibited ERK phosphorylation induced by TNF-α in HaCaT cells. These results indicate that the SBF-improving effects of EECP may be associated with ERK inhibition.

In patients with atopic dermatitis, the concentration of ICAM-1 in the blood is elevated compared with that in healthy individuals, and keratinocytes also exhibit increased expression of ICAM-1 [40]. In our results, EECP effectively suppressed TNF-α–induced ICAM-1 mRNA overexpression. These findings suggest that the inhibitory effect of EECP on immune cell infiltration involves not only the suppression of MCP-1, a chemokine, but also the downregulation of ICAM-1 expression.

Regarding papain, one of the key enzymes found in papaya peel, previous studies have demonstrated that it alleviates atopic skin inflammation through multiple mechanisms. Specifically, papain treatment reduces TEWL, downregulates AD-related cytokines such as Th2 and Th17, decreases epidermal thickness and mast cell numbers, and consequently lowers serum IgE levels. In HaCaT keratinocytes, papain also significantly inhibits TNF-α/IFN-γ–induced IκBα phosphorylation and NF-κB activation, exhibiting strong anti-inflammatory activity. Collectively, these effects help stabilize key symptoms of dermatitis, including pruritus, erythema, inflammation, and hyperkeratosis [23]. These findings suggest that papain may play a key role in the anti-dermatitic effects of EECP observed in the present study.

While corticosteroids and histamine antagonists remain common treatments for CD, the long-term use of glucocorticoids is associated with many serious adverse effects, including body weight reduction and skin atrophy [16,17]. The DEX-treated group in this study showed significant body weight loss, whereas no such reduction was observed in the EECP-treated groups (Appendix A).

In wild-type mice, DEX reduced paw edema by approximately 70–95%, whereas in MKP-1-deficient mice, it showed little to no anti-edematous effect [41]. This indicates that DEX alleviates allergic inflammation through an MKP-1–dependent mechanism. Glucocorticoids such as DEX bind to the glucocorticoid receptor (GR) and translocate into the nucleus, where they regulate the expression of thousands of genes. Because GRs are present in almost all cells and tissues, GR-related compounds often cause undesired side effects such as osteoporosis, diabetes, and hypertension [42]. In contrast, the anti-inflammatory constituents of papaya such as papain, flavonoids, vitamin C, and carotenoids have not been reported to exert their effects through MKP-1 activation or GR binding. For these reasons, adverse effects commonly observed with glucocorticoids, such as a reduction in body and spleen weight, seem to be absent in EECP treatment.

Inflammatory skin diseases can lead to systemic inflammation accompanied by immune cell infiltration into spleen and splenomegaly [43]. Interestingly, a significant increase in the spleen body weight ratio was observed in the DNFB control group in this study (Appendix A). In contrast, the DEX group showed a significant decrease in spleen body weight ratio compared to the control group. Since the spleen is a major target of glucocorticoid-induced catabolism [44], and spleen weight in the DEX group was lower than that in the NOR group, this observation indicates potential suppression of systemic immunity. In contrast, EECP did not significantly reduce spleen weight, suggesting that it may not induce general immune deficiency and exert its therapeutic effects by targeting local inflammation.

Glucocorticoids are known to suppress keratinocyte proliferation and inhibit differentiation in the epidermis, resulting in reductions in skin thickness and weight [45]. In this study, animals in the DEX-treated group exhibited a significant decrease in skin weight, whereas no such reduction was observed in the EECP-treated groups (Figure 1C). While this may be interpreted as a pharmacological effect of DEX, it may also be considered an adverse outcome, given the skin-thinning side effects of glucocorticoids.

To assess the effects of EECP on normal skin, an additional study was conducted. The experimental procedures were identical to those of the main study, except for the absence of DNFB treatment. The results indicated that six days of topical EECP application did not induce any noticeable changes in skin surface appearance, skin color, thickness, or weight. Moreover, topical application of EECP for 6 days significantly increased skin moisture content and WHC compared with the normal group treated with vehicle only (Appendix A). Previous studies have also reported that topical formulations containing papaya fruit extract, such as soaps, lotions, and lipsticks, did not cause skin irritation or sensitization [46]. In addition, a study using body lotion formulations containing young papaya skin (peel) extract found that both physical stability tests and irritation tests yielded “Nil” for irritation [47]. Taken together, although the available data are limited, these findings suggest that EECP, as used in the present study, appears to be relatively safe compared with glucocorticoids.

## 4. Materials and Methods

### 4.1. Preparation of the Papaya Peel Extract

Papaya fruit (Green papaya, Dole, Polomolok, Philippines) was purchased through an internet vendor (Coupang, Seoul, 
Republic of Korea). Peel was obtained after washing and removing the flesh, and then dried in a forced convection oven 
(JSOF-150, JSR, Gongju, Republic of Korea) at 60 °C for 24 hrs. Extractions were performed using a standard 
laboratory procedure [16]. Briefly, dried papaya peel (100 g) was 
soaked in 500 mL of 70% ethanol, sonicated for 5 min, and extracted for 24 h. The supernatant was then collected, and 
the peel was subjected to a second extraction using an additional 500 mL of 70% ethanol for another 24 h after 5 min of 
sonication. The extract was then filtered through Whatman No. 20 filter paper, concentrated using a rotary evaporator 
(Eyela, Tokyo, Japan), and freeze-dried (Labconco, Kansas City, MO, USA). This process yielded 9.31 g of lyophilized 
extract (9.31% yield). A sample of this ethanol extract of *C. papaya* peel (EECP, voucher No. MS2022-1029) was 
stored at the Division of Pharmacology, School of Korean Medicine, Pusan National University (Appendix A).

### 4.2. Animals

Animal experiments were performed using 7-week-old male Balb/c mice obtained from Hana Biotech (Hwaseong, Republic of Korea). Animals were maintained in a specific pathogen-free (SPF) environment under a controlled 12 h light/dark cycle with free access to standard laboratory chow and water. All experimental protocols were reviewed and approved by the Institutional Animal Care and Use Committee (IACUC) of Pusan National University (Approval No. PNU-2022-0208; 8 August 2022), in accordance with institutional and national guidelines.

### 4.3. Induction of CD and the Experimental Design

Animals were randomly divided into six groups: the treatment-naive group (NOR, *n* = 5), the CD control group (CD CTL, *n* = 8), three EECP treatment groups (60, 180, and 600 µg/day; *n* = 8 per group), and the dexamethasone-treated positive control group (DEX, *n* = 8). All animals, except those in the NOR group, were sensitized by applying 10 μL of DNFB (0.1%, *v*/*v*) dissolved in AOO (acetone:olive oil, 4:1) to both ears for three consecutive days. On day 5, dorsal hair was removed, and DNFB (0.1%, *v*/*v*; 40 μL/day) was applied to the dorsal skin on days 8, 10, 12, and 14 to induce CD. Three different doses of EECP (1, 3, or 10 mg/mL; 60 μL/day) or DEX (2.5 mg/mL; 60 μL/day) were applied to shaved backs once daily from day 9 to day 14. EECP and DEX were dissolved in ethanol and then diluted in AOO (EAOO; ethanol:AOO, 4:1). The summary of the experimental design is provided in Appendix A.

### 4.4. Skin Observation and Assessment of Skin Lesions and Weight

Photographs were taken of the shaved dorsal skin of all mice using a digital camera (IXUS 990 IS, Canon, Oita, Japan) on day 15. Skin lesion severity (roughness, excoriation, scabs, and erythema) was assessed using a standardized 4-point grading scale, ranging from 0 (absent) to 3 (severe), and total lesion scores were calculated by summation. On day 15, skin samples (5 mm diameter) were collected using a biopsy punch and weighed using a microbalance (Sartorius AG, Göttingen, Germany).

### 4.5. Assessment of Skin Color

Skin color was assessed on day 15. Erythema and melanin levels were measured at three different sites per mouse using a skin colorimeter (DSM II, Cortex Technology, Horsens, Denmark), and skin color was calculated by averaging these measurements.

### 4.6. Skin Water Content and Water-Holding Capacity

Skin water content and WHC were evaluated using a skin hygrometer (Scalar Corporation, Tokyo, Japan) at 22.6 °C and 43% relative humidity before sacrifice (on day 15). Briefly, skin water contents were assessed at three dorsal locations per mouse. WHC values were determined by measuring skin water content four times at 30 s intervals. The first measurement was taken immediately after removing a wet gauze (1 × 1 cm, soaked in distilled water) that had been placed on the shaved dorsal skin for 30 s.

### 4.7. Histopathological Examination

Inflamed tissue samples were excised, fixed, embedded in paraffin, and sectioned for hematoxylin and eosin (H&E) staining. The stained tissue slides were observed for histological changes under a light microscope (Carl Zeiss AG, Oberkochen, Germany) at ×100.

### 4.8. Quantitative Analysis of Epidermal Thickness and Immune Cell Infiltration

Epidermal hyperplasia was assessed by measuring the vertical distance from the basal lamina to the outer layer. Three randomly selected areas were analyzed per slide using Zen imaging software (Zen 3.0, ZEISS, Jena, Germany), and three measurements were taken per area. Epidermal thicknesses were obtained by averaging these values. Immune cell infiltration was quantified using a counting grid in three randomly selected, non-overlapping fields per slide.

### 4.9. Cytokine Levels

Cytokine concentrations in dorsal skin samples were quantified using a Mouse Inflammation Cytometric Bead Array Kit (BD Biosciences, San Jose, CA, USA) [16]. Excised dorsal skin tissues were homogenized in a protein extraction buffer (Thermo Scientific, Mount Prospect, IL, USA), and 50 μg samples of lysates were used to determine TNF-α, IFN-γ, IL-6, and MCP-1 levels. All experimental procedures were conducted according to manufacturers’ instructions.

### 4.10. Cell Culture

HaCaT cells, an immortalized human keratinocyte, were purchased from CLS Cell Lines Service GmbH (Eppelheim, Germany) and cultured in Dulbecco’s Modified Eagle’s Medium (DMEM; HyClone™, Logan, UT, USA) supplemented with 10% fetal bovine serum (FBS; Corning, Corning, NY, USA) and 1% penicillin–streptomycin (Gibco; Thermo Fisher Scientific, Waltham, MA, USA). Cells were maintained at 37 °C in a 5% CO_2_/air atmosphere, detached using 0.05% trypsin–EDTA at 80–90% confluence, and subcultured in complete growth medium. For experiments, cells were seeded in a 6-well plate at 3 × 10^5^ cells/well, incubated overnight, and then treated with 10 ng/mL of TNF-α.

### 4.11. Protein Extraction and Western Blot Analysis

HaCaT cells were seeded in 100 mm culture dishes at a density of 1 × 10^6^ cells/well and incubated overnight to allow cell attachment. The cells were pre-treated with EECP for 6 h, followed by stimulation with TNF-α (10 ng/mL). The control group was treated with the solvent vehicle under the same conditions. For TNF-α stimulation, cells were treated for 15 min for the analysis of MAPK pathway activation and for 24 h for the evaluation of filaggrin expression. Following protein extraction using Pro-Prep protein extraction solution (iNtRON Biotechnology, Daejeon, Republic of Korea), protein concentration was determined using the Bradford assay. For Western blot analysis, 30 μg of total protein per lane was loaded onto an SDS-PAGE gel and separated by electrophoresis. The proteins were then transferred to a PVDF membrane at 320 mA for 1.5 h, and immunoblots were visualized using a luminescent analyzer (Amersham™ Imager 600, GE Healthcare, Amersham, UK).

### 4.12. Total RNA Isolation and Quantitative Real-Time PCR

HaCaT cells were seeded in 12-well plates at a density of 2 × 10^5^ cells per well and incubated overnight to allow attachment. Cells were pre-treated with EECP for 1 h prior to stimulation with TNF-α (10 ng/mL) for 2 h. Control cells were treated with the solvent vehicle under the same conditions. Total RNA was isolated from HaCaT cells using TRIzol reagent (Invitrogen; Thermo Fisher Scientific, Carlsbad, CA, USA). Briefly, an M-MLV cDNA synthesis kit (Enzynomics, Daejeon, Republic of Korea) was used to prepare cDNA from 2 μg of total RNA. Quantitative real-time PCR was performed with the TOPreal SYBR Green qPCR premix (Enzynomics, Daejeon, Republic of Korea) on a Rotor-Gene Q system (Qiagen, Hilden, Germany). The primer sets used are shown in Appendix A. Target gene expressions were normalized versus GAPDH.

### 4.13. Statistical Analysis

All in vivo experiments were conducted once, and each group, except for the NOR group, consisted of *n* = 8 mice. All in vitro experiments were independently repeated three times, with *n* = 3 technical replicates per experiment. Statistical analysis was performed using one-way ANOVA followed by Dunnett’s post hoc test to assess intergroup differences. All analyses were conducted with GraphPad Prism version 5.01 for Windows (GraphPad Software Inc., La Jolla, CA, USA). Data are presented as mean ± standard deviation (SD), and statistical significance was defined as *p* < 0.05.

## 5. Conclusions

In this study, EECP demonstrated therapeutic efficacy by alleviating CD symptoms, including scabbing, fissures, roughness, and erythema, and enhancing skin water content and holding capacity. In addition, topical EECP attenuated histopathological abnormalities and significantly suppressed the production of TNF-α, IFN-γ, IL-6, and MCP-1, which are key inflammatory mediators of the pathogenesis of CD. Furthermore, this cytokine suppression reduced inflammatory responses and subsequent immune cell infiltration. These anti-inflammatory effects ultimately improved clinical presentation and skin barrier function. Collectively, these findings suggest that EECP has potential use as a plant-derived therapeutic alternative with minimal adverse effects.

## Figures and Tables

**Figure 1 ijms-26-11122-f001:**
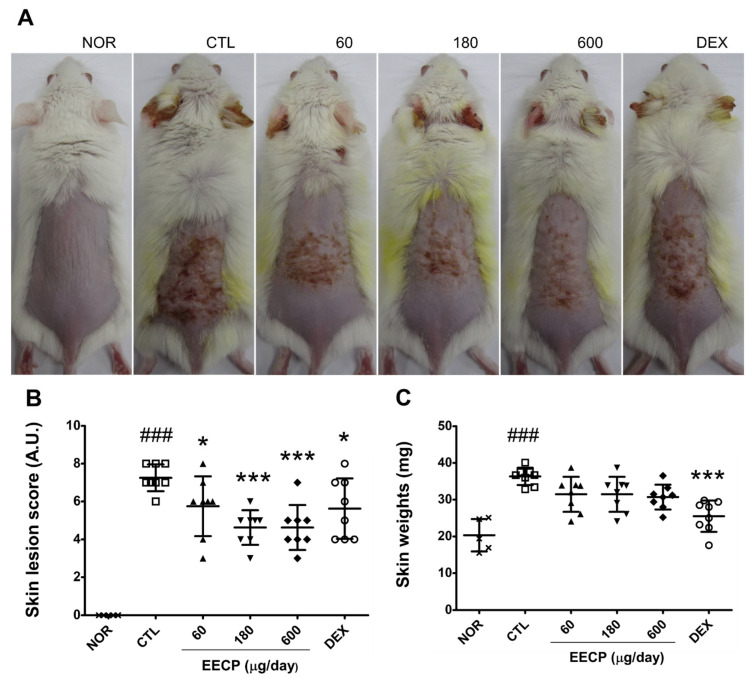
The effect of EECP on skin lesions and skin weights in CD mice. Skin surface (**A**) was observed and skin weight (**C**) was measured on day 15. The skin lesion score was assessed using a 4-point scale (**B**). NOR, non-treated normal mice; CTL, non-treated CD mice; 60, 60 μg/day of EECP; 180, 180 μg/day of EECP; 600, 600 μg/day of EECP; DEX, 150 μg/day of dexamethasone. A.U., arbitrary units; EECP, ethanol extract of *C*. *papaya* peel. Values are expressed as means ± SDs. ^###^ *p* < 0.001 vs. NOR, * *p* < 0.05 and *** *p* < 0.001 vs. CTL.

**Figure 2 ijms-26-11122-f002:**
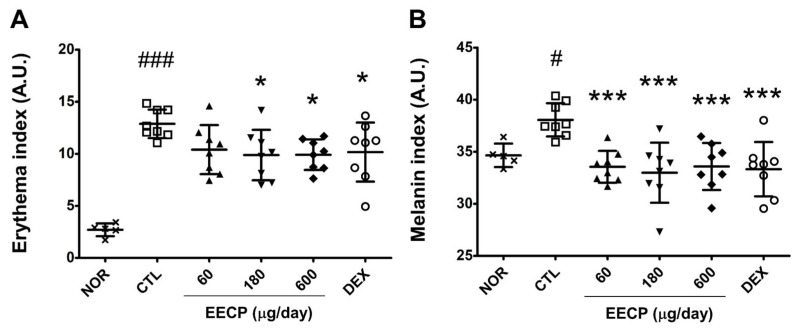
The effects of EECP on erythema and melanin indices. Erythema (**A**) and melanin indices (**B**) were quantified using a skin colorimeter. A.U., arbitrary units; EECP, ethanol extract of *C*. *papaya* peel. Values are expressed as means ± SDs. *^###^ p* < 0.001 and *^#^ p* < 0.05 vs. NOR, * *p* < 0.05 and *** *p* < 0.001 vs. CTL.

**Figure 3 ijms-26-11122-f003:**
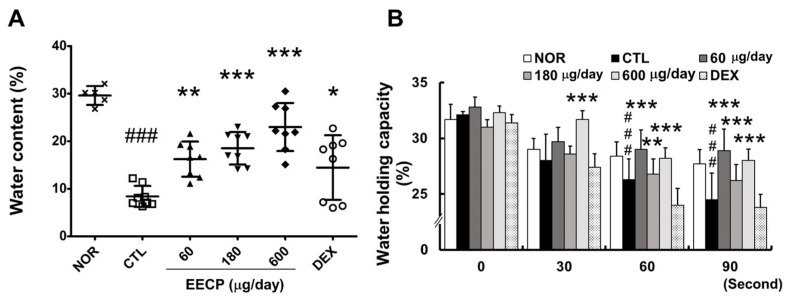
The effects of EECP on skin water content and WHC in CD mice. (**A**) Water content; (**B**) water-holding capacity. EECP, ethanol extract of *C*. *papaya* peel. Values are expressed as means ± SDs. ^###^ *p* < 0.001 vs. NOR, * *p* < 0.05, ** *p* < 0.01 and *** *p* < 0.001 vs. CTL.

**Figure 4 ijms-26-11122-f004:**
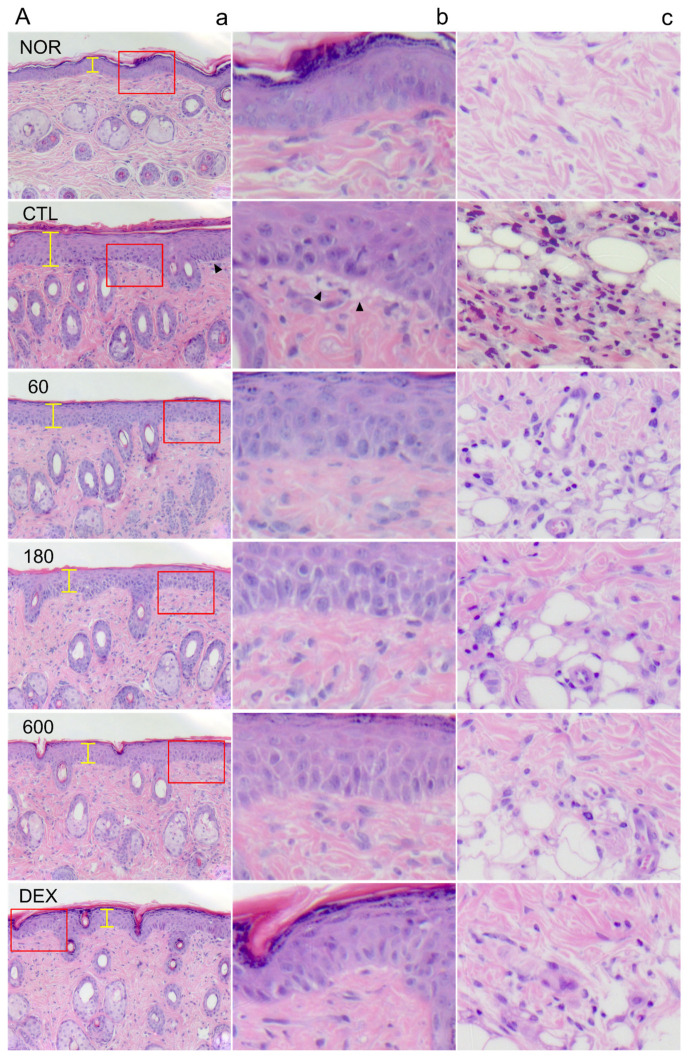
The effects of EECP on histopathological abnormalities in CD mice. (**A**) (**a**), the overall condition of the epidermis and dermis (100×); (**b**), basal layer (400×); (**c**), infiltrated immune cells in dermis (400×). The red squares shown in panel (**a**) are magnified in panel (**b**). Yellow bars indicate the thickness of the epidermis and solid wedges denote regions of spongiotic changes. (**B**) epithelial thickness; (**C**) number of infiltrated immune cells. EECP, ethanol extract of *C. papaya* peel. Abbreviations are consistent with those in Figure 1. Values are expressed as means ± SDs. ^###^ *p* < 0.001 vs. NOR, * *p* < 0.05, ** *p* < 0.01 and *** *p* < 0.001 vs. CTL.

**Figure 5 ijms-26-11122-f005:**
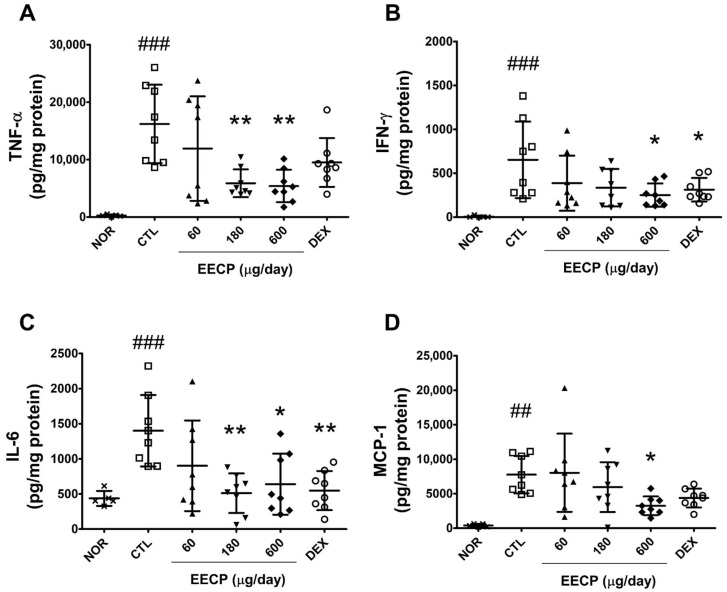
The effects of EECP on levels of TNF-α, IFN-γ, IL-6, and MCP-1 in inflamed tissues. (**A**) TNF-α; (**B**) IFN-γ; (**C**) IL-6; (**D**) MCP-1. EECP, ethanol extract of *C. papaya* peel. Values are expressed as means ± SDs. ^##^ *p* < 0.01 and ^###^ *p* < 0.001 vs. NOR, * *p* < 0.05 and ** *p* < 0.01 vs. CTL.

**Figure 6 ijms-26-11122-f006:**
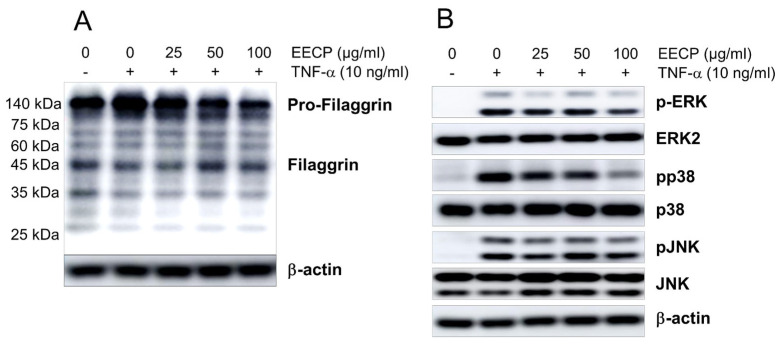
The effects of EECP on pro-filaggrin, filaggrin, and MAPK (Mitogen-Activated Protein Kinase) signaling pathways in HaCaT cells. Pro- filaggrin, filaggrin (**A**), and MAPK signaling molecules (**B**) were determined using Western blots. EECP, ethanol extract of *C. papaya* peel.

**Figure 7 ijms-26-11122-f007:**
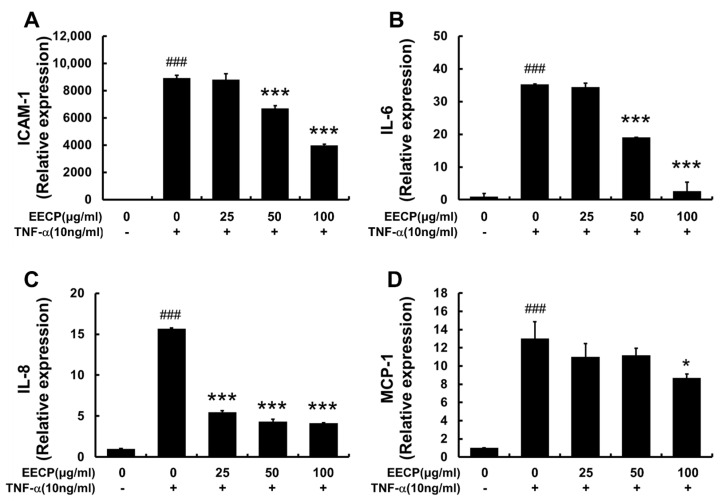
The effects of EECP on TNF-α induced increases in ICAM-1 and cytokine expression in HaCaT cells. Relative mRNA expressions of ICAM-1 (**A**), IL-6 (**B**), IL-8 (**C**) and MCP-1 (**D**) were determined using Q-PCR (quantitative real-time Polymerase Chain Reaction). EECP, ethanol extract of *C. papaya* peel. Values are expressed as means ± SDs of three independent experiments. ^###^ *p* < 0.001 vs. non treated group, * *p* < 0.05 and *** *p* < 0.001 vs. TNF-α treated group.

## Data Availability

The data presented in this study are available upon request from the corresponding author.

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
