# Peer review of "The Anti-Inflammatory and Skin Barrier Function Recovery Effects of Carica papaya Peel in Mice with Contact Dermatitis"

_ijms, 2025, doi:10.3390/ijms262211122_

Round 1
Reviewer 1 Report
Comments and Suggestions for Authors
In their manuscript „The Anti-inflammatory and Skin Barrier Function Recovery Effects of Carica papaya Peel in Mice with Contact Dermatitis,” Park et al. describe the consequences of papaya peel application onto the mouse skin in a model of DNFB-induced contact dermatitis and to TNF-α-activated HaCat cells. They observed a dose-dependent reduction in skin inflammation by ethanol extract of C. papaya peel (EECP)as assessed by parameters for skin integrity and appearance, as well as the expression of pro-inflammatory cytokines. Additionally, the authors showed that EECP resulted in reduced inflammatory response of stimulated HaCat cells.
This outcome is not unexpected, as several studies have been published reporting on the anti-inflammatory effect of papaya extracts, in particular for those papaya components like papain, vitamin C, phenols, and flavonoids (reviewed e.g. by Akanda et al., 2025, doi.org/10.1007/s10787-025-01780-4; Babalola et al., 2024, doi.org/10.1016/j.arabjc.2023.105369; Kong et al., 2021, doi.org/10.3390/biology10040287), which were also detected in the peel as mentioned by the authors (Mukta et al., 2025). The results of this study thus confirm what was already known, without, however, examining in detail the respective contributions of the papaya peel components listed.
Apart from that, the study remains largely descriptive without providing mechanistic explanations of how the anti-inflammatory effect of EECP occurs.
Major concerns:
1. The authors do not explain why they chose this set-up, but the regimen of the DNFB application reflects the model of allergic contact dermatitis, mainly mediated by T cells. However, the T cell response is not addressed at all; the authors only describe invading immune cells by histology. However, this cannot be seen at the magnification and resolution shown in their figure (Fig.4). The authors could present a detailed analysis of the cell populations using flow cytometry or at least detect characteristic markers immunohistochemically.
2. Although not clearly stated all analyses seems to be done on day 15. By switching the sensitization from ears to back skin with challenge onto the ears, the progression of ear swelling could be measured, which not only provides a parameter for the induced inflammation that is easy to quantify, but also allows the attenuating effect of papaya extract to be recorded over time.
3. A description of the solvent used for the EECP is lacking. It is also not specified in what order DNFB and EECP were applied, and if the solvent without extract was also applied to the skin of the animals that only received DNFB. There is also no control to check whether EECP has an effect on otherwise untreated skin without CD.
4. There is no indication of how often the experiments (in vivo and in vitro) were repeated.
5. The analysis of skin integrity in vivo was performed by measuring the water content and water holding capacity of the skin. It is not specified at which time point/day these analyses were carried out. Since protein isolation was performed from the skin, the authors could also have examined filaggrin and other structural skin components.
6. The in vitro experiments also lack information on when and for how long the cells were treated with EECP and whether the controls received the solvent.
7. The authors compare the anti-inflammatory effect of EECP with that of Dex and conclude that different mechanisms of action are involved, since no body weight loss was observed under the treatment with EECP in contrast to Dex (p9, l237-239). However, this finding is also not further addressed and is only mentioned in the discussion
8. In their conclusion, the authors suggest “that EECP has potential use as a plant-derived therapeutic alternative with minimal adverse effects“ (p11, l370-371). What are these minimal adverse effects? No tests have been conducted on skin compatibility or possible toxic effects of EECP in this study.
Author Response
Thank you for your detailed and scholarly review. I have carefully considered all of your comments, and the revised sections have been marked in red.

Reviewer 2 Report
Comments and Suggestions for Authors
The research requires an introduction and discussion of the most important compounds in the hat, such as the enzyme papain, which helps regenerate cells and treat some skin diseases.
Second: Vitamin C, which increases collagen.
Third: Flavonoid compounds such as quercetin, myricetin, and beigenin, which have antioxidant and anti-inflammatory roles.
Fourth: Carotenoids reduce radiation damage.
Fifth: The mechanism of these compounds can be used to reduce inflammation.
Comments on the Quality of English Language
The English could be improved to more clearly express the research.
Author Response

(The authors gave the same response as above.)

Reviewer 3 Report
Comments and Suggestions for Authors
The manuscript "The Anti-inflammatory and Skin Barrier Function Recovery Effects of Carica papaya Peel in Mice with Contact Dermatitis" is devoted to a relevant topic. However, it evokes an ambiguous impression. On the one hand, the authors conducted a significant volume of experiments and obtained meaningful results possessing elements of scientific novelty. At the same time, Section 3 Discussion is written as if detached from Section 2 Results. Lines 256-262 stand completely apart from the main concept of the Discussion.
In the presented manuscript, Section 3 Discussion is largely devoted to describing factors associated with skin diseases, while the description of the effects of papain/Carica papaya extract is given only a few sentences. In my opinion, a more successful solution would be to devote the Discussion section to describing the impact of various synthetic and natural medicines on scabbing, fissures, roughness, and erythema, and skin water content, etc., as well as to describing their side effects, which would potentially be absent in the papaya extract.
Thus, the article's concept is not bad, but the authors should align the Introduction, Results, and Discussion sections in such a way that a single idea is traced through them – Carica papaya extract peel demonstrated therapeutic potential by effectively suppressing inflammation and restoring skin barrier function. These findings support the potential use of Carica papaya extract as a safe and effective botanical candidate for the treatment of contact dermatitis and the promotion of overall skin health with a minimal number of side effects compared to synthetic medicines.
Author Response

(The authors gave the same response as above.)

Round 2
Reviewer 1 Report
Comments and Suggestions for Authors
The revised manuscript shows significant improvements, but I still have a few more points of criticism.
Comments to the point-by-point response:
Point 2: As already noted, it is not possible to discern the stated cell infiltration and the difference between the various treatments from the histological images shown in Fig. 4A. Nor is it possible to recognize spongiosis in the basal layer described on pages 4, l113-114. The authors should be able to provide images with better resolution and implement section with higher magnification in which the changes/differences they describe are visible.
Point 6: The question of how often the experiments were performed is not fully answered. The authors should make a clear statement (e.g. in the statistic section), how often the experiments were conducted, and how many animals/replicates were used per individual experiment. The authors only added a comment to Fig.7
General comments:
The individual results should be better linked to each other, which would also improve the readability of the manuscript. For example, explain why individual experiments were performed and why they are relevant in the context of allergic contact dermatitis (e.g why were HaCat cells stimulated with TNF and not DNFB).
Presentations of the data should be done with individual values instead of filled bars.
The use of upper and lower case letters in parallel in Fig. 1 and Fig. 4 is confusing, especially since they are not explained in Fig.4. The authors could, for example, label the respective pictures according to the labeling of their bar charts (NOR/CTL/EECP 60/....) instead of using lower case letters.
In Fig.6 B the authors show pJNK and JNK which is not mentioned in the text.
Author Response
We have carefully reviewed the reviewers’ comments, and the changes made in the second round are highlighted in blue.

Reviewer 3 Report
Comments and Suggestions for Authors
The authors took my comments into account and significantly revised the manuscript. I recommend the article for publication.
Author Response
We have carefully reviewed the reviewers’ comments, and the changes made in the second round are highlighted in blue.
Point 1: The authors took my comments into account and significantly revised the manuscript. I recommend the article for publication.
Response 1: We sincerely thank you for your generous and positive evaluation of our manuscript, despite its shortcomings.
Other corrections:
- Added information regarding the abbreviation (A.U.) (L90-91).
- Added information regarding the abbreviation (EECP) (L151, 159).
- Other minor grammatical corrections.
Round 3
Reviewer 1 Report
Comments and Suggestions for Authors
In the revised version, the authors have addressed all points of criticism in a largely satisfactory manner, thereby significantly improving their manuscript.